# The Characteristics of the Favard E4 Glenoid Morphology in Cuff Tear Arthropathy: A CT Study

**DOI:** 10.3390/jcm9113704

**Published:** 2020-11-18

**Authors:** Gilles Walch, Philippe Collotte, Patric Raiss, George S. Athwal, Marc Olivier Gauci

**Affiliations:** 1Centre Orthopédique Santy, Hôpital Privé Jean Mermoz, Ramsay Générale de Santé, 69008 Lyon, France; gilleswalch15@gmail.com (G.W.); docteurcollotte@gmail.com (P.C.); 2OCM Clinic, Steinerstrasse 6, 81369 Munich, Germany; 3Roth/McFarlane Hand and Upper Limb Center, St Joseph’s Health Care, Western University, London, ON N6G 4E7, Canada; gathwal@uwo.ca; 4Institut Universitaire Locomoteur & Sport, Unité de Recherche Clinique Côte d’Azur (UR2CA), Hôpital Pasteur 2, Université Côte d’Azur, 06000 Nice, France; marcoliviergauci@gmail.com

**Keywords:** shoulder, cuff tear arthropathy, glenoid anteversion, humeral head subluxation, Favard E4, reverse shoulder arthroplasty

## Abstract

Background: Cuff tear arthropathy (CTA) is characterized by superior migration of the humeral head with superior erosion of the glenoid. Rarely, humeral head migration can be anteroinferior with associated anterior erosion of the glenoid, a pattern described by Favard as the type E4 glenoid. The purpose of this retrospective imaging study was to analyze the 2D and 3D characteristics of the E4 glenoid. Methods: A shoulder arthroplasty database of 258 cuff tear arthropathies was examined to identify patients with an E4 type deformity. This resulted in a study cohort of 15 females and 2 males with an average age of 75 years. All patients had radiographs and CT scans available for analysis. CT-scan DICOM (Digital Imaging and Communications in Medicine) data were uploaded to a validated three-dimensional (3D) imaging software. Muscle fatty infiltration, glenoid measurements (anteversion, inclination), and humeral head subluxation according to the scapular plane were determined. Results: The mean anteversion and inclination of the E4 cohort were 32° ± 14° and −5° ± 2, respectively. The mean anterior subluxation was 19% ± 16%. All cases had severe grade 3 or 4 fatty infiltration of the infraspinatus, whereas only 65% had grade 3 or 4 subscapularis fatty infiltration. A significant correlation existed between glenoid anteversion and humeral head subluxation (*p* < 0.001), but no correlation was found with muscle fatty infiltration. The CT analysis demonstrated an acquired erosive biconcave morphology in 11 patients (65%) and monoconcavity in 6 patients (35%). Conclusion: The E4 type glenoid deformity in cuff tear arthropathy is characterized by an anterior erosion and anteversion associated with anterior subluxation of the humeral head.

## 1. Introduction

Anteroinferior glenoid erosion in rotator cuff tear arthropathy (CTA) has been described as the type E4 glenoid morphology by Favard in 2000 [1] and reported in the scientific literature by Sirveaux et al. [2] and Lévigne in 2011 [3] (Figure 1a,b).

This type of glenoid erosion, which is predominantly located in the inferior part of the glenoid, was present in 3% of the 461 shoulders analyzed in a multicenter study on CTA [3]. The main feature of the type E4 glenoid morphology was the inferior inclination of the glenoid, which leads to inferior tilting (varus position) of the glenoid component and a significantly lower incidence of scapular notching (25% vs. 83% notch incidence in the type E2 glenoid morphology *p* = 0.004) [3]. The unique aspects of the type E4 are the anteroinferior erosion, glenoid anteversion, and static anteroinferior humeral head subluxation, which is in contrast to typical cuff tear arthropathy findings of posterosuperior erosion and superior subluxation (Favard types 2 and 3). Unfortunately, the literature pertaining to the Favard E4 morphology is limited. There is no detailed information available regarding the pathology itself as well as the percentage in patients with cuff tear arthropathy. As such, the goal of this retrospective imaging study was to analyze the two-dimensional (2D) and the three-dimensional (3D) characteristics of the E4 type glenoid in cuff tear arthropathy.

## 2. Methods

### 2.1. Study Cohort

The shoulder arthroplasty databases from three institutions (Lyon and Nice, France; London; and ON, Canada) were used to identify patients with a Favard type E4 morphology. The radiographs and computer-tomographic (CT) scans of the patients were retrospectively analyzed by two experienced shoulder surgeons with more than 10 years of experience with shoulder replacement surgery (M.O.G. and G.W.). Patients with primary osteoarthritis, previous arthroplasty surgery, previous cuff repair, infection, instability, rotator cuff tear without arthritis, inflammatory arthritis, and post-traumatic arthritis were excluded. This resulted in 258 patients with a primary diagnosis of rotator cuff tear arthropathy (CTA) (Figure 2).

In this CTA cohort, glenoid morphology was analyzed and classified according to the Favard Classification [1]. The type E4 glenoid morphology was defined as glenoid anteversion and anterior subluxation of the humeral head with respect to the scapular plane. A threshold value of less than 45% humeral head subluxation was used to define anterior subluxation and less than 0° of glenoid retroversion to describe glenoid anteversion [4,5,6,7,8,9,10]. This resulted in the identification of 17 patients, which represented 6.6% of the entire CTA group (95% CI = 4.2 − 10.4). Both examiners agreed with the diagnosis of an E4 glenoid in those 17 cases. All patients in the E4 study cohort had shoulder radiographs, computed tomography scans, and DICOM (Digital Imaging and Communications in Medicine) images suitable for analysis with an imaging software system (Glenosys; Imascap, Wright Medical, Memphis, TN, USA). The study cohort consisted of 15 females and 2 males. There were 13 (77%) right shoulders and the mean age was 75 years (range, 54 to 84).

### 2.2. Imaging Analysis

Each patient’s shoulder CT-scan DICOM data were uploaded to a validated 3D imaging software program (Glenosys; Imascap, Brest, France) [11,12,13,14]. The DICOM data were segmented into 3D images of the scapula and humeral head. The methods for 3D measurements of version and inclination using Glenosys software have been reported previously [11]. The glenoid version angle was computed as the angulation between the scapular plane and the glenoid best-fit sphere centerline projected on the transverse scapular plane. The inclination was measured on the basis of the transverse line that runs through the base of the supraspinatus fossa between the trigonum scapulae and the middle of the glenoid vault. This line was algorithmically created by the software program. The inclination angle was defined as the angle between the transverse line and the glenoid centerline projected on the scapular plane. The humeral head subluxation index was calculated by dividing the 3D volumetric portion of the humeral head posterior to the scapular plane by the whole volume of the humeral head (Figure 3a,b).

This 3D measurement was performed automatically by the software program, and it eliminates the variability in 2D axial slice selection and measurements. A humeral head subluxation index threshold value of 45% for anterior subluxation was used as published previously [4,5,6,7,10,15]. Goutallier’s classification was used to assess muscle fatty infiltration on CT images [16]. Standard shoulder radiographs were used to examine the position of the humeral head with respect to the glenoid and scapula.

### 2.3. Statistics

The values for the quantitative variables were expressed as means and range values. The values for the qualitative variables were expressed as a number of patients per group, frequency (%). Comparisons were done using the Fisher exact or chi-squared test for categorical variables and the paired Student’s test or the Wilcoxon rank-signed test for quantitative variables. Correlations were assessed using the Spearman or Pearson correlation coefficient for quantitative variables. Data were normally distributed. All statistical analyses were performed with SAS^®^ 9.4 (SAS, Cary, CA, USA). The significance was evaluated by calculating the *p*-value and was set at *p* < 0.05.

## 3. Results

For the E4 study cohort (Table 1), the average anteversion was 32° ± 14° (range, 4° to 58° anteverted), inclination was on average inferior −5° ± 2° (range, −43° to +31°) and the average subluxation was 19% ± 16% (range, 0% to 44%). Reported degrees of glenoid retroversion and inclination in cases without degenerative diseases vary between 0° and 10° in the literature [4].

Fatty infiltration of the rotator cuff muscles was severe as seen in Table 2.

The entire E4 cohort had severe grade 3 or 4 fatty infiltration of the infraspinatus muscle, whereas only 65% had severe infiltration (>2) of the subscapularis. The CT-based glenoid morphology demonstrated a biconcave erosion in 11 patients (65%) and a monoconcave appearance in 6 patients (35%) (Figure 4a,b).

In the biconcave group, the neoglenoid was located inferiorly in four cases and involved the entire anterior glenoid face from superior to inferior in seven cases. Among the six patients with a monoconcave erosion, the erosion was directed anteriorly and the humeral head was not contacting the acromion; however, humeral head anterior contact with the coracoid was present in 4 (67%) cases. When examining the humeral head in the study cohort, five patients had radiographic and CT findings consistent with partial avascular necrosis, and interestingly no Hill–Sachs lesions were identified. Radiographically, the humeral head in the E4 cohort was consistently anteroinferiorly subluxated (Figure 5).

### Correlations

There were no statistically significant differences in the glenoid measurements of anteversion, inclination, and subluxation according to age, gender, side, and muscle fatty infiltration. A statistical correlation was identified between anteversion and subluxation (*p* < 0.001); as glenoid anteversion increased, the severity of anterior humeral head subluxation also increased (Figure 6).

Additionally, there was a weaker correlation between anteversion and inclination (*p* = 0.019), and no correlation between inclination and subluxation (*p* = 0.220). The E4 mono or biconcave appearance on CT scan was not correlated with any of the 3D measurements of anteversion, inclination, or subluxation nor with muscle fatty infiltration.

## 4. Discussion

There is limited literature defining and characterizing the E4 glenoid morphology. Initially, the definition was descriptive based on radiographs demonstrating anteroinferior glenoid erosion and anteroinferior humeral head subluxation occurring in rotator cuff tear arthropathy. This E4 pathoanatomy is in contrast to the typical CTA pattern of proximal humeral head migration with associated posterosuperior glenoid erosion and contact of the humeral head with the undersurface of the acromion. The etiology of the typical CTA pattern is attributed to posterosuperior rotator cuff deficiency with severe fatty infiltration of the supraspinatus and infraspinatus muscles with or without teres minor involvement, which results in superior migration of the humeral head due to the deltoid [17]. The superiorly migrated humeral head then articulates with the undersurface of the acromion and the superior aspect of the glenoid with resultant erosions in this area.

The E4 morphology is relatively rare, as only 17 cases were identified from a selected group of 258 patients with cuff tear arthropathy. Interestingly, 88% of cases were women, so this glenoid pathology seems to be more prominent in females. The 3D measurements confirmed that E4 patients have consistently anteverted glenoids with a minimum value of 4° and a maximum value of 58°. This anteversion can be related to anterior erosion of the glenoid, which may involve the entire glenoid face resulting in a monoconcave appearance on CT scan or biconcave with the eroded aspect being the most anterior portion of the glenoid. We found no statistical difference regarding age, gender, and side involved between the monoconcave and biconcave patterns. Similarly, we found no statistical differences in the degrees of anteversion (*p* = 0.560), subluxation (*p* = 0.975), or inclination (*p* = 0.972) between monoconcave and biconcave morphologies. The atypical bone erosion in E4 glenoids might be challenging for the treatment with reverse shoulder arthroplasty. As the bone stock of the scapula in those cases is limited, fixation of the baseplate can be difficult, and a careful preoperative planning seems to be crucial.

Severe fatty infiltration of the infraspinatus muscle was present in all our cases; however, we cannot attribute the E4 erosion pattern exclusively to this finding, as classic CTA also has high degrees of infraspinatus involvement. Initially, we believed that the E4 pattern could be related to subscapularis deficiency; however, in our cohort, high-grade subscapularis fatty infiltration was only present in 11 cases (65%) without any statistical correlation with anteversion or subluxation values. Therefore, with the number available in our study, we can only conclude that a muscular imbalance between internal and external rotators cannot fully explain the E4 pattern and that the etiology is likely multifactorial. The severity of anterior subluxation is statistically correlated with the glenoid anteversion (*p* < 0.0001) but not with the CT scan appearance of monoconcavity or biconcavity (*p* = 0.975). Additionally, our study cannot prove causation, as we cannot determine whether the anterior erosion is the cause of subluxation or whether the anterior subluxation preceded the erosion. To investigate the potential of a traumatic etiology for the E4, all available clinical charts available were checked; no patients reported any type of substantial trauma or previous episode of instability that could explain the atypical pattern of the E4. Furthermore, we excluded all patients with a previous surgery that could have influenced the erosion or subluxation pattern.

Despite the anterior subluxation or dislocation of the humeral head in our patients, we did not observe any Hill–Sachs lesions of the humeral head as seen in traumatic anterior shoulder instability or locked anterior dislocations. The erosion in the E4 is consistently on the glenoid side; therefore, the absence of a Hill–Sachs defect is a distinguishing feature of the E4, allowing differentiation from the locked anterior dislocation. Matsoukis et al. reported on 11 patients with fixed anterior dislocations treated with an anatomic Total-Shoulder-Arthroplasty (TSA) [18]. The authors reported seven complications in five patients, and four of the seven complications were recurrent anterior dislocations [18]. Werner et al. and Raiss et al. both reported on a series of locked anterior shoulder dislocations in older patients [19,20]. Although both series reported high degrees of anterior glenoid bone loss and in some cases massive rotator cuff tears, the mean age of the patients was younger (71 years old in both series) than in our series (75 years old). Moreover, most of the patients in the Werner et al. and Raiss et al. series recalled a clear traumatic event that resulted in the anterior dislocation, a finding that we did not observe in our E4 cohort. All the patients in the Werner et al. and Raiss et al. studies were treated with a reverse total shoulder arthroplasty. Interestingly, the rate of failure of the glenoid component was high, with Werner et al. reporting a 10% failure rate and Raiss et al. having a 22% failure rate. This higher-than-anticipated failure rate of the glenoid component underscores the challenges in managing patients with anterior glenoid bone loss and anterior instability.

In Neer’s 1983 article on cuff tear arthropathy, the author reported on some cases of anterior or posterior chronic dislocations as part of his CTA cohort. It is conceivable that some of the instability cases reported by Neer in the CTA paper were actually a type E4. Overall, the type E4 glenoid morphology is a rare presentation of cuff tear arthropathy characterized by anteversion, severe fatty infiltration of the supraspinatus and infraspinatus muscles, and anteroinferior subluxation of the humeral head. Although the subscapularis muscle displayed severe fatty infiltration in 65% of the patients, it does not seem to be a key causative factor in the development of the E4 glenoid. However, it is possible that the subscapularis may have been torn in the remaining 35% of patients, as this cannot be definitely diagnosed on CT scans.

The clinical relevance of the type E4 is difficult to distinguish from the typical CTA patterns of superior migration seen in the E2 and E3 types. Saying this, the E4 morphology may mislead surgeons into believing that the posterosuperior rotator cuff is intact, as typically there is an increased acromohumeral distance (Figure 5). However, in our series, posterosuperior cuff involvement with the E4 was 100%. Additionally, with the E4 there was a 65% rate of high-grade fatty infiltration of the subscapularis. This high rate of associated subscapularis rupture may result in a lower subscapularis repair rate, potentially decreasing internal rotation strength or adversely affecting implant stability. Overall, similar to the Favard E2 and E3, the E4 is best managed with reverse shoulder arthroplasty. Similarly, the placement of the glenoid baseplate requires particular attention to correct version and inclination, which can be addressed by eccentric reaming as well as metallic or bony augmentation.

This study does have limitations. It is a retrospective database study with a small number of cases. Additionally, magnetic resonance imaging, which may be better at identifying rotator cuff tears, was not available or not done in our patient cohort.

## 5. Conclusions

The type E4 glenoid deformity occurs in cuff tear arthropathy and is characterized by anterior glenoid erosion, anteversion, and anteroinferior humeral head subluxation. The associated findings include a 100% rate of high-grade fatty infiltration in the posterosuperior cuff and a 65% rate in the subscapularis muscle.

## Figures and Tables

**Figure 1 jcm-09-03704-f001:**
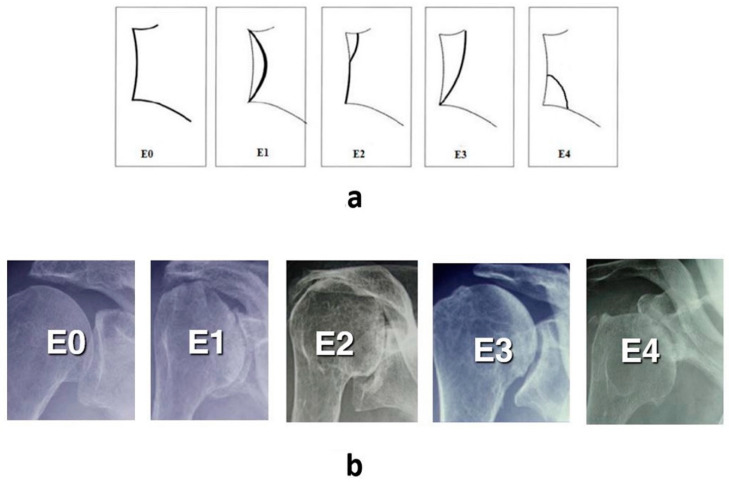
Favard et al.’s classification of the different types of glenoid erosion depicted schematically (**a**) and radiographically (**b**). The classification progresses from E0 to E4. E0: superior humeral head migration with no glenoid erosion; E1: concentric medialized glenoid erosion; E2: glenoid erosion predominantly in the superior pole; E3: global glenoid erosion more severe in the superior pole; E4: glenoid erosion predominantly in the anteroinferior pole.

**Figure 2 jcm-09-03704-f002:**
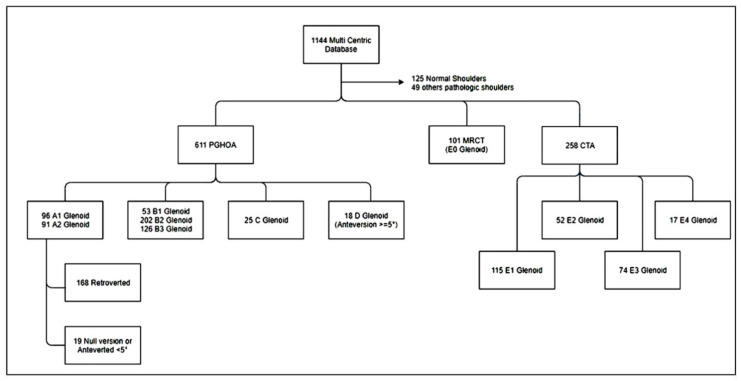
A flow chart of our database. Among a series of 1144 computer-tomographic (CT) scans, there were 258 cuff tear arthropathy (CTA) cases. The CTA group was divided in to 115 E1 glenoids, 52 E2 glenoids, 74 E3 glenoids, and 17 type E4 glenoids (6.5% of all CTA cases). primary gleno-humeral osteoarthritis (PGHOA): E0. E1-E4 corresponds to CTA (Cuff-Tear-Arthropathy), E0 corresponds to MRCT (Massive-Rotator-Cuff-Tears).

**Figure 3 jcm-09-03704-f003:**
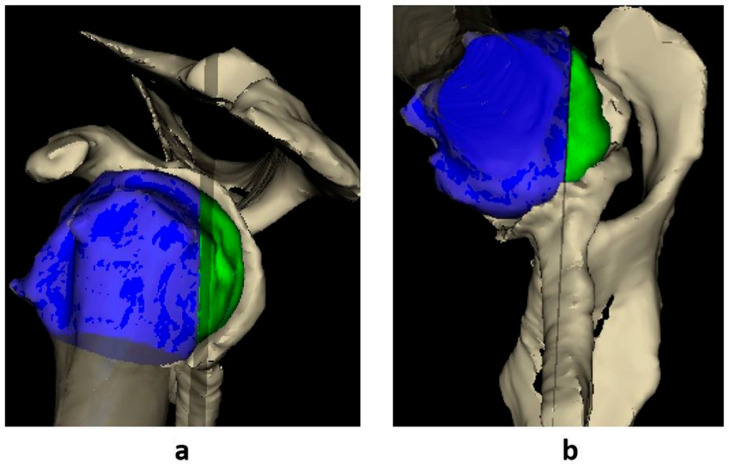
Humeral head subluxation index was calculated by dividing the three-dimensional (3D) volumetric portion of the humeral head posterior to the scapular plane (green) by the whole volume of the humeral head. A sagittal view (**a**) of an E4 glenoid in a left shoulder with the black plane representing the scapular plane. The glenohumeral joint from an inferior vantage point (**b**) demonstrates 11% of the humeral head is posterior to the scapular plane (11% subluxation).

**Figure 4 jcm-09-03704-f004:**
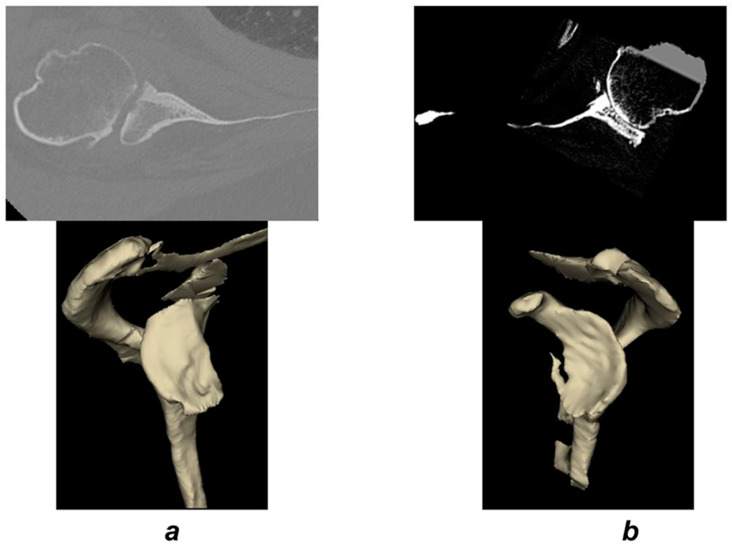
The CT scan appearance was either biconcave (**a**) or uniconcave (**b**).

**Figure 5 jcm-09-03704-f005:**
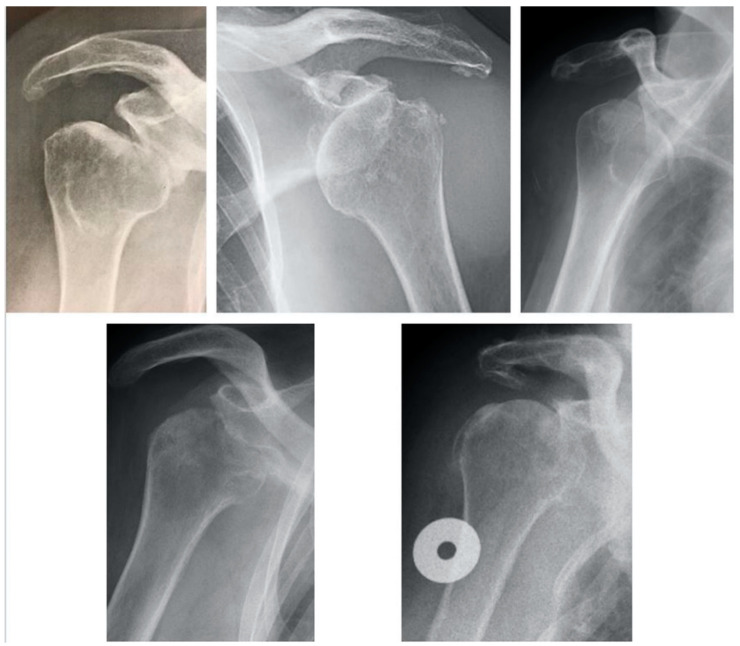
Several anteroposterior radiographs demonstrating the anterior or anteroinferior humeral head subluxation typical of the E4 glenoid morphology as described by Favard et al. [2].

**Figure 6 jcm-09-03704-f006:**
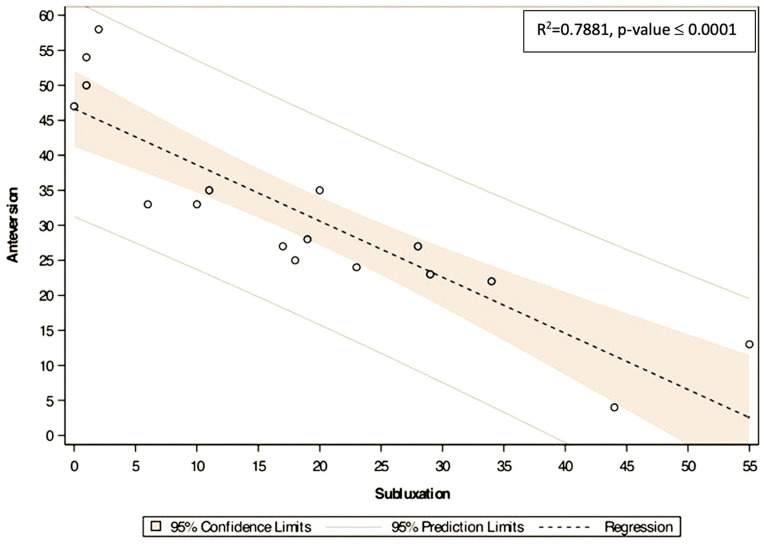
Linear statistical correlation between subluxation (*x*-axis) and anteversion (*y*-axis) (*p* < 0.0001) indicating that when anteversion increased so did anterior subluxation.

**Table 1 jcm-09-03704-t001:** Study cohort of 17 patients with demographics, glenoid indices, and imaging findings.

Patient Number	Gender	Age	Side	Anteversion	Inclination	Subluxation	FI SubScap	FI SupraS	FI InfraS	FI Teres Minor	CT Scan Morphology	Coracoid Contact	X-ray Appearance
1	F	66	R	22	0	34	1	4	4	4	biconcave	-	Ant Inf sublux
2	M	80	L	35	−7	11	4	4	4	4	monocave	-	Ant Inf sublux
3	F	80	L	23	−19	29	4	4	4	0	monocave	+	Ant Inf sublux
4	F	83	R	54	−35	1	3	4	4	0	biconcave	-	Ant Inf sublux
5	F	80	R	28	7	19	3	3	3	0	biconcave	-	Ant Inf sublux
6	F	78	R	27	2	28	4	4	4	0	biconcave	-	Ant Inf sublux
7	F	74	L	27	31	17	4	4	4	0	monocave	+	Ant Inf sublux
8	F	70	R	35	−2	20	4	3	4	0	biconcave	-	Ant Inf sublux
9	F	84	R	24	−6	23	1	2	3	0	monocave	-	Ant Inf sublux
10	F	83	R	47	−13	0	2	4	4	0	biconcave	-	Ant Inf sublux
11	M	54	L	58	−43	2	1	4	4	0	monocave	+	Ant Inf sublux
12	F	76	R	4	14	44	1	1	4	0	monocave	-	Ant Inf sublux
13	F	75	R	13	−7	44	2	4	4	0	biconcave	-	Ant Inf sublux
14	F	75	R	25	−16	18	4	4	4	0	biconcave	+	Ant Inf sublux
15	F	72	R	33	18	10	4	4	4	0	biconcave	-	Ant Inf sublux
16	F	75	R	33	−2	6	4	4	4	0	monocave	+	Ant Inf sublux
17	F	73	R	50	−3	1	4	4	4	0	biconcave	-	Ant Inf sublux

FI: fatty infiltration, SubScap: subscapularis, SupraS: supraspinatus, InfraS: infraspinatus, R: right, L: left.

**Table 2 jcm-09-03704-t002:** Distribution of fatty infiltration of rotator cuff muscles.

	SubScap	Supraspinatus	Infraspinatus	Teres Minor
Stage 4	9	13	15	2
Stage 3	2	2	2	-
Stage 2	2	1	-	-
Stage 1	4	1	-	-
Stage 0	-	-	-	15

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
