# Peer review of "The Characteristics of the Favard E4 Glenoid Morphology in Cuff Tear Arthropathy: A CT Study"

_jcm, 2020, doi:10.3390/jcm9113704_

Round 1

Reviewer 1 Report

Dear Authors,

Congratulations for your work on an interesting topic and for your large and deep expertise.

I have some questions and suggestions in order to clarify the background, the rationale and results of your study.

Do you think it coul be better to change the title? "The characteristics of the Farvard E4 glenoid morphology in cuff tear arthropathy: a CT study"

Abstract:

Line 16: "retrospective clinical": please clarify in the methods what is clinical in your study.

To clarify the aim of your study, I think that it could be better to use the same words of the Introduction. "...the 2-D and the 3-D..."

Introduction:

It could be better to clarify what is unknown in this setting and how and why we should fill the gap?

Discussion/Conclusions:

In my opinion, if you clarify the rationale of the present study, it could be easier to undestand how your results fill this gap and what does this mean for us going forward.

Author Response

Congratulations for your work on an interesting topic and for your large and deep expertise.

Thanks.

I have some questions and suggestions in order to clarify the background, the rationale and results of your study.

Do you think it coul be better to change the title? "The characteristics of the Farvard E4 glenoid morphology in cuff tear arthropathy: a CT study"

Thanks for this advice. This was changed accordingly.

Abstract:

Line 16: "retrospective clinical": please clarify in the methods what is clinical in your study.

This is correct and was deleted as required.

To clarify the aim of your study, I think that it could be better to use the same words of the Introduction. "...the 2-D and the 3-D..."

This was changed accordingly.

Introduction:

It could be better to clarify what is unknown in this setting and how and why we should fill the gap?

A sentence was included from line 52-54 as follows:

"There is no detailed information available regarding the pathology itself as well as the percentage in patients with cuff-tear arthropathy."

Discussion/Conclusions:

In my opinion, if you clarify the rationale of the present study, it could be easier to undestand how your results fill this gap and what does this mean for us going forward.

This was adressed accordingly in the discussion (line 167-170) as follows:

"The atypical bone erosion in E4 glenoids might be challenging for the treatment with reverse shoulder arthroplasty. As the bone stock of the scapula in those cases is limited fixation of the baseplate can be difficult and a careful preoperative planning seems to be crucial."

Reviewer 2 Report

Reviewer comments Walch et al 2020

General comments:

This paper give some characteristics on an uncommon type of cuff tear arthropathy and the paucity in the literature on this subtype Favard E4 makes these these results interesting. The aim was to analyze characteristics and this has been done for some aspects. There are, however, some questions and comments.

  • The authors consistently use the name “Favard et al” classification, referring to a book-chapter from 2001, which is hard to find in full-text or even in abstract format. In other literature on classification of cuff tear arthropathy, for example in a CTA review by Kappe et al (JSES 2011), the “Favard classification” refers to another classification (Favard L, Lautmann S, Clement P. Osteoarthritis with massive rotator cuff tear: The militations of its current definitions. In: Gazielly D, Gleyze P, Thomas T, editors. The cuff. Paris: Elsevier; 1997. p. 261-5.). In the Kappe review, the Sirveuax classification (Sirveaux F, Favard L et al. J Bone Joint Surg Br 2004; 86:388-95.) uses the glenoid for classification in E-types, just like in this paper. This is potentially highly confusing for the reader and could be either corrected or at least clarified in the text.

Abstract

  • Line 17: says 248 CTA cases. This must be wrong. The rest of the manuscript uses 258 CTA cases.

Introduction

  • Line 44-45: The apparently largest study of CTA images found 3% E4 out of 461 shoulders (i.e 14 cases; could be written out). The paper at hand found 6.5% (17 out of 258= 6.6%, should be corrected). Since there, as expected, seems to be different estimates of the frequency of E4 in different samples, it would be appropriate in this paper to give a confidence interval for the proportion of cases found, which in this case probably would go from 4% to 10%. This tells the reader that with 95% likelihood the frequency of E4 among patients with CTA will be between 4% and 10%.

Methods

  • Line 59: “analyzed by 2 experienced surgeons”. Experience=more than 10 years of experience? Did they use consensus discussion for definitive diagnosis? Did the 2 assessors agree in all 17 cases that these were in fact E4? Were there other potential candidates that were discussed? Did they go through all 1144 cases in the database (numbers taken from figure 2, could be pointed out in the text)? Agreement statistics for the 2 assessors?
  • Line 69: Authors defined E4 as having “glenoid anteversion and anterior subluxation“, but only one of these characteristics is defined: “less than 45% humeral head subluxation”. What was the definition of anteversion? This could be related to the normal position of the glenoid (according to literature with some measure of spread). From Figure 1, the reader is given the impression that the classification of E-type is done on a frontal projection. How can the anterior component of subluxation be assessed on a frontal projection? If proper classification of E4 only can be made with x-ray and CT, this could be pointed out in the discussion as a limitation.
  • Line 71: To help the reader to understand the 45% definition: What percentage is found in normal shoulders?

Statistics

  • Both parametric and nonparametric methods are mentioned, but no method of evaluation of whether the data was normally distributed or not are shown.

Results

  • Line 112 +: Is it possible to give the readers the values found in normal shoulders (from the literature or own data)?
  • Line 114, Table 1: 88% of the cohort were females (15 out of 17). The results are therefore valid primarily for women, something which could be pointed out in the discussion. This finding also give rise to a hypothesis (that E4 may be more common in women), which could be explored in future studies and mentioned in the discussion.
  • Line 117: “Table 2. Discussion”. What discussion?

   Supraspinatus and infraspinatus are usually written in one word.

  • Line 138+: the actual correlation coefficients and their 95% CI should be given instead of just p-values!

Discussion

  • Line 163: “We found no statistical difference regarding age, gender, side involved between the monoconcave and biconcave patterns. Similarly, we found no statistical differences in the degrees of anteversion (p=0.560), subluxation (p=0.975), or inclination (p=0.972) between monoconcave and biconcave morphologies.”. These are results and should be reported in the results section!
  • Line 167-174. Interesting discussion. It thus seems as classic CTA and E4 cases all have incompetent infraspinatus muscles. Do the authors have data on compensatory hypertrophy of the teres minor in these 17 cases? Since we don´t know the integrity of the subscapularis tendon in the 17 cases in present study, E4 cases may have a stronger posterior side compared to the anterior side, and thus hypothetically be more prone to anterior subluxation and anteroinferior glenoid erosion.
  • Line 178: “we carefully reviewed the clinical history of the patient cohort” is a method and should be reported in the methods section. How was this careful review performed? All medical charts back in time or just the note from the orthopedic surgeon meeting the patient?
  • Could these 17 cases have had anteroinferior dislocations? The matter is discussed but, in previous studies referred to, only Raiss et al report of Hill-Sachs impressions in all their patients. Interestingly Raiss et al also had 7 out of 23 who did not recall a fall or trauma. Could these 17 cases be patients who simply cannot remember a trauma, but who have had it none the less?

Author Response

General comments: 

This paper give some characteristics on an uncommon type of cuff tear arthropathy and the paucity in the literature on this subtype Favard E4 makes these these results interesting. The aim was to analyze characteristics and this has been done for some aspects. There are, however, some questions and comments.

  • The authors consistently use the name “Favard et al” classification, referring to a book-chapter from 2001, which is hard to find in full-text or even in abstract format. In other literature on classification of cuff tear arthropathy, for example in a CTA review by Kappe et al (JSES 2011), the “Favard classification” refers to another classification(Favard L, Lautmann S, Clement P. Osteoarthritis with massive rotator cuff tear: The militations of its current definitions. In: Gazielly D, Gleyze P, Thomas T, editors. The cuff. Paris: Elsevier; 1997. p. 261-5.). In the Kappe review, the Sirveuax classification (Sirveaux F, Favard L et al. J Bone Joint Surg Br 2004; 86:388-95.) uses the glenoid for classification in E-types, just like in this paper. This is potentially highly confusing for the reader and could be either corrected or at least clarified in the text.

Thanks for this very good point. The reference we added was the first time this classification was mentioned. As you mentioned correctly, we added the reference 20 (20. Sirveaux F, Favard L, Oudet D, Huquet D, Walch G, Molé D Grammont inverted total shoulder arthroplasty in the treatment of glenohumeral osteoarthritis with massive rupture of the cuff. Results of a multicentre study of 80 shoulders. .J Bone Joint Surg Br. 2004 Apr;86(3):388-95. doi: 10.1302/0301-620x.86b3.14024.) to avoid confusion.

Abstract

  • Line 17: says 248 CTA cases. This must be wrong. The rest of the manuscript uses 258 CTA cases.

Sorry this was a typo. 258 cases are correct and was changed in line 17.

Introduction

  • Line 44-45: The apparently largest study of CTA images found 3% E4 out of 461 shoulders (i.e 14 cases; could be written out). The paper at hand found 5%(17 out of 258= 6.6%, should be corrected).

This was corrected in line 75.

  • Since there, as expected, seems to be different estimates of the frequency of E4 in different samples, it would be appropriate in this paper to give a confidence interval for the proportion of cases found, which in this case probably would go from 4% to 10%. This tells the reader that with 95% likelihood the frequency of E4 among patients with CTA will be between 4% and 10%.

This is absolutely correct and was embedded in the text in line 75 (95% CI 4.2-10.4).

Methods

  • Line 59: “analyzed by 2 experienced surgeons”. Experience=more than 10 years of experience? Did they use consensus discussion for definitive diagnosis? Did the 2 assessors agree in all 17 cases that these were in fact E4? Were there other potential candidates that were discussed? Did they go through all 1144 cases in the database (numbers taken from figure 2, could be pointed out in the text)? Agreement statistics for the 2 assessors?

Both analyzers had more than 10 years of experience in shoulder replacement surgery (embedded in line 60-61). As the morphology of the E4 glenoid is completely different compared to the E0-E3 glenoids there were no other potential candidates discussed. Both analyzers reviewed all 258 CTA cases.

  • Line 69: Authors defined E4 as having “glenoid anteversion and anterior subluxation“, but only one of these characteristics is defined: “less than 45% humeral head subluxation”. What was the definition of anteversion? This could be related to the normal position of the glenoid (according to literature with some measure of spread).

Glenoid anteversion was defined as less than 0° of retroversion. This information was embedded in line 73-74.

  • From Figure 1, the reader is given the impression that the classification of E-type is done on a frontal projection. How can the anterior component of subluxation be assessed on a frontal projection? If proper classification of E4 only can be made with x-ray and CT, this could be pointed out in the discussion as a limitation.

Historically, the Favard classification was established by using ap radiographs. In the current study we used additionally 2D and 3D CT examinations in order to better understand the pathology itself and to help surgeons treating these glenoids. We therefore do not believe that this is a limitation.

  • Line 71: To help the reader to understand the 45% definition: What percentage is found in normal shoulders?

To our knowledge, the normal distribution in 3D measurement has not yet been described. In theory, the perfect centered head would have a 50% subluxation (meaning no subluxation). We are not sure if this information helps the reader or if it may lead to confusing. We kindly ask the Editor if this should be embedded.

Statistics

  • Both parametric and nonparametric methods are mentioned, but no method of evaluation of whether the data was normally distributed or not are shown.

Data were normally distributed. This was embedded in line 111-112.

Results

  • Line 112 +: Is it possible to give the readers the values found in normal shoulders (from the literature or own data)?

Very good point. This was embedded in line 117-118: with reference 1: Reported degrees of glenoid retroversion and inclination in cases without degenerative diseases varies between 0-10° in literature1.

  • Line 114, Table 1: 88% of the cohort were females (15 out of 17). The results are therefore valid primarily for women, something which could be pointed out in the discussion. This finding also give rise to a hypothesis (that E4 may be more common in women), which could be explored in future studies and mentioned in the discussion.

This is a very good point. We added this in the discussion in line 164-165.

  • Line 117: “Table 2. Discussion”. What discussion?

   Supraspinatus and infraspinatus are usually written in one word.

Sorry. This was a typo. It was corrected to “Distribution of fatty infiltration of rotator cuff muscles” and Supraspinatus & Infraspinatus were written in one word.

  • Line 138+: the actual correlation coefficients and their 95% CIshould be given instead of just p-values!

This is correct, however, CI were given in Fig. 6. We did not want to duplicate the results and therefore we did not include them again. We ask the Editor if we should mention them again.

Discussion

  • Line 163: “We found no statistical difference regarding age, gender, side involved between the monoconcave and biconcave patterns. Similarly, we found no statistical differences in the degrees of anteversion (p=0.560), subluxation (p=0.975), or inclination (p=0.972) between monoconcave and biconcave morphologies.”. These are results and should be reported in the results section!

These results were already described in the Results section (line 141-151). We mentioned those again in order to help the reader to better follow the discussion at this part. If needed, we can delete this paragraph of course.

  • Line 167-174. Interesting discussion. It thus seems as classic CTA and E4 cases all have incompetent infraspinatus muscles. Do the authors have data on compensatory hypertrophy of the teres minor in these 17 cases? Since we don´t know the integrity of the subscapularis tendon in the 17 cases in present study, E4 cases may have a stronger posterior side compared to the anterior side, and thus hypothetically be more prone to anterior subluxation and anteroinferior glenoid erosion.

Thanks for this good point. We reviewed the 17 cases again but we did not find signs for hypertrophy in our patients.

  • Line 178: “we carefully reviewed the clinical history of the patient cohort” is a method and should be reported in the methods section. How was this careful review performed? All medical charts back in time or just the note from the orthopedic surgeon meeting the patient?

We changed this according to your recommendation in order to avoid confusing. This was changed to: “…all available clinical charts available were checked in line 187-188…”

  • Could these 17 cases have had anteroinferior dislocations? The matter is discussed but, in previous studies referred to, only Raiss et al report of Hill-Sachs impressions in all their patients. Interestingly Raiss et al also had 7 out of 23 who did not recall a fall or trauma. Could these 17 cases be patients who simply cannot remember a trauma, but who have had it none the less?

Referring to your comment above, we checked all available charts and we found no history of trauma/ shoulder dislocation in these 17 cases. However, as this is a retrospective study we cannot exclude that.